**Data Availability Statement:** The research was based on the analysis of anonymized data that can be publicly accessed through the Kidney Stone Registry (http://kidneystoneregistry.com.s3-

# Stone decision engine accurately predicts stone removal and treatment complications for shock wave lithotripsy and laser ureterorenoscopy patients

**Peter A. Noble** [1] *, **Blake D. Hamilton**[2], **Glenn Gerber**[3]

**1** Department of Microbiology, University of Alabama Birmingham, Birmingham, AL, United States of America, **2** School of Medicine, University of Utah, Salt Lake City, UT, United States of America, **3** University of Chicago Medical Center, Chicago, IL, United States of America

\* panoble2017@gmail.com

## Abstract

Kidney stones form when mineral salts crystallize in the urinary tract. While most stones exit the body in the urine stream, some can block the ureteropelvic junction or ureters, leading to severe lower back pain, blood in the urine, vomiting, and painful urination. Imaging technologies, such as X-rays or ureterorenoscopy (URS), are typically used to detect kidney stones. Subsequently, these stones are fragmented into smaller pieces using shock wave lithotripsy (SWL) or laser URS. Both treatments yield subtly different patient outcomes. To predict successful stone removal and complication outcomes, Artificial Neural Network models were trained on 15,126 SWL and 2,116 URS patient records. These records include patient metrics like Body Mass Index and age, as well as treatment outcomes obtained using various medical instruments and healthcare professionals. Due to the low number of outcome failures in the data (e.g., treatment complications), Nearest Neighbor and Synthetic Minority Oversampling Technique (SMOTE) models were implemented to improve prediction accuracies. To reduce noise in the predictions, ensemble modeling was employed. The average prediction accuracies based on Confusion Matrices for SWL stone removal and treatment complications were 84.8% and 95.0%, respectively, while those for URS were 89.0% and 92.2%, respectively. The average prediction accuracies for SWL based on Area-Under-the-Curve were 74.7% and 62.9%, respectively, while those for URS were 77.2% and 78.9%, respectively. Taken together, the approach yielded moderate to high accurate predictions, regardless of treatment or outcome. These models were incorporated into a Stone Decision Engine web application (http://peteranoble.com/webapps.html) that suggests the best interventions to healthcare providers based on individual patient metrics.

## Introduction

The incidence and prevalence of kidney stones in people is increasing globally presumably due to dietary practices and global warming [1]. In the United States, about 11% of the population

website-us-west-2.amazonaws.com/) and/or email contact: info@trans-stat.com.

**Funding:** The author(s) received no specific funding for this work.

**Competing interests:** NO authors have competing interests

**Abbreviations:** ANN, Artificial Neural Network; AUC, Area-Under-the-Curve; BMI, Body Mass Index; DCD2, Dornier Compact Delta II; DCD3, Dornier Compact Delta III; DCS, Dornier Compact Sigma; DMH30, Dornier Medilas H30; DMH35, Dornier Medilas H35; SWL, Extracorporeal shock wave lithotripsy; LV100, Lumenis Versapulse 100 watt; LV20, Lumenis Versapulse 20 watt; NIH, National Institute of Health; OC30, Odyssey Convergent 30 watts; SDE, Stone Decision Engine; SF2, Storz F2; SMOTE, Synthetic Minority Oversampling Technique; SSLXT, Storz SLX-T; URS, Ureterorenoscopy.

will have kidney stones in their lifetime [2]. The increasing incidence of kidney stones presents a dilemma to healthcare professionals because the 'optimal' intervention to remove the stones varies by approach [3], patient health, age, preference, and body size [4–8], stone size and composition [9], and stone location [10].

Two interventions most often used to remove/fragment stones, include: shock wave lithotripsy (SWL) and laser ureterorenoscopy (URS). SWL uses high-energy shock waves to fragment stones into small particles that eventually pass out of the body in urine [11]. This intervention is a less invasive than URS but not as effective in terms of attaining stone-free status–that is, patients might require additional treatments [12]. A laser attached to the URS is used to fragment stones, which are subsequently either transported out of the body in the urine stream or removed during the procedure [13]. Two drawbacks of URS are: higher incidence of treatment complications and more costly, sometimes requiring longer hospital stays than patients treated by SWL [14, 15]. A survey of intervention decisions suggests most patients prefer SWL to URS [16] and a recent Evidence Review by NIH states only 'small benefits of URS over SWL'—yet clinical and cost effectiveness favor SWL [17]. Selecting the 'optimal' intervention for patients is therefore not straightforward; an approach that helps healthcare professionals with these decisions is highly desired.

Artificial neural network (ANN) models are computational systems or algorithms designed to simulate human intelligence and perform tasks that typically require human intelligence. These models learn from data and experience, enabling them to make predictions, recognize patterns, and solve problems without being explicitly programmed for each specific task. They are now widely used in urology to detect kidney stones in videos [18] and images [19–24], predict sepsis risk [25, 26] and lithotripsy treatment outcomes [27–29], and set SWL machine parameters [30].

The objective of this study was to build a Stone Decision Engine (SDE) based on mining a database containing information on previous interventions (SWL and URS). The databases include information on patient metrics (such as age and Body Mass Index (BMI)), stone removal successes/failures, and evidence of treatment complications. We determined the prediction probabilities for various treatment outcomes based on these metrics and the uncertainty of the predictions by repeated independent statistical analyses. ANN models were used to find patterns in the 17242 patient records. The equations of forty models were extracted and incorporated into a SDE application that healthcare professionals can use in patient counseling to predict SWL or URS outcomes based on patient metrics.

## Materials and methods

### Ethics statement

The research relied on the analysis of anonymized data accessible through the Kidney Stone Registry. The anonymous dataset lacks identifiable information, ensuring no possible linkage to personal data.

### Electronic medical data

The database consisted of 80,000+ patients who had undergone SWL or URS treatments at multiple sites throughout the United States. We selected 20,000 patient records between February 19th 2018 and August 31st, 2021. We then excluded records with missing or erroneous data to end up with 17242 patient records. Individual patient consent was not required as no patient identifiable records were used in the study.

A variety of SWL and URS instruments were used to treat patients. Specifically, SWL was performed using the Dornier Compact Delta II (DCD2), Dornier Compact Delta III (DCD3),

Dornier Compact Sigma (DCS) (Weßling, Germany), Storz F2 (SF2), or Storz SLX-T (SSLXT) instruments by experienced physicians. Laser URS was performed using Dornier Medilas H20 DMH20, Dornier Medilas H30 (DMH30), Dornier Medilas H35 (DMH35), Lumenis Versapulse 100 watt (LV100) (San Jose, CA), Lumenis Versapulse 20 watt (LV20), or Odyssey Convergent 30 watt (OC30) (Alameda, CA) instruments by experienced physicians.

## Coding of variables in the data sets

**SWL data.** Label, coding, units: Anticoagulants used prior to treatment (True: 0, False: 1), DCD2 (True: 1, False: 0), DCD3 (True: 1, False: 0), DCS (True: 1, False: 0), SF2 (True: 1, False: 0), SSLXT (True: 1, False: 0), Stone location in ureters (True: 1, False: 0), Stone location in kidney (True: 1, False: 0), Stone not specifically located in the kidney or ureters (0ther location) (True: 1, False: 0), Sex (Male: 1, Female: 0), Body Mass Index (BMI, kg/m$^2$), Age of the patient at time of the procedure (years), Stone width (mm), Stone length (mm), Stone side (Left: 0, Right,1), Other medical conditions (e.g., Diabetes or other without diabetes, True: 1, False: 0).

**URS data.** Label, coding, units: Sex (Male: 1, Female: 0), DMH20 (True: 1, False: 0), DMH30 (True: 1, False: 0), DMH35 (True: 1, False: 0), LV100 (True: 1, False: 0), LV20 (True: 1, False: 0), OC30 (True: 1, False: 0), Age of the patient at time of the procedure (years), BMI (kg/m$^2$).

**Target outcomes.** Two definitions of stone removal outcomes were used: (i) 'stone free' or stone fragments < 4 mm were assigned a value of '0', and (ii) stone fragments > 4mm or 'no change in stone size' were assigned a value of '1'. These outcomes were determined by a physician's review of the follow-up X-ray images and confirmed with patient records indicating no further treatment was required. There were two definitions of 'treatment complications': (i) a patient with 'no complication' was assigned a value of '0', and (ii) a patient with a treatment complication was assigned a value of '1'. Typical treatment complications included pain, fever, urinary tract infection, hematoma, post-operational bleeding, "*steinstrasse*", prolonged dysuria, ureteral perforation, burning, hydronephrosis, acute kidney injury, tachycardia, prolonged gross hematuria, and obstructing fragments.

**Standardization of the data.** Prior to building the ANN models, continuous variables were standardized by their corresponding minimum (min) and maximum (max) with the formula:

standardized variable = (raw variable–variable min)/ (variable max–variable min)

## ANN modeling

The data sets were randomly split into 70% training, 15% testing, and 15% validation. The architecture of the ANN models consisted of an input, a hidden, and an output layer. The number of neurons in input layer was dependent on the number of input variables. The optimal number of neurons in the hidden layer was empirically determined by selecting a range of numbers (e.g., the square root of the number of inputs to the actual number of inputs) and assessing model accuracy using a Confusion Matrix (i.e., (True positives + True negatives)/ (False Positives + False Negatives + True Positives + True Negatives). The output layer consisted of a single neuron, the target variable (i.e., stone removal success or treatment complication). In some cases, the model accuracy was assessed by including all data (i.e., training, testing and validation data sets) into the Confusion Matrix, while in others, only the combined testing/validation data sets were used, as specified in the Results section below. The Neuroet package [31] settings used for training were as follows: scaling method, standard linear function (0, 1); transfer function for input and output neurons, Log-Sigmoid; training method, Levenberg-Marquardt. Training was automatically stopped when the global error between

outputs and targets was minimized after several iterations. Weights and biases were retained to build the final equations in MS Excel and C++ programs.

**Balanced and SMOTED data.** Preliminary studies showed that the ANN models had difficulties in learning the decision boundaries due to severe imbalances of the data. For example, more patient records had successful stone removals than unsuccessful ones and even fewer patient records involved treatment complications. To address this issue, two data augmentation approaches were used: (i) balancing the training data set with equal number of records for each group (i.e., equal number of successes and failures), and (ii) increasing the number of records in the minority class by synthesizing data using Synthetic Minority Oversampling Technique (SMOTE) [32].

The balanced data set approach involved randomly selecting $x$ number of records from the majority class to make them equal in number to those in the minority class. The SMOTE approach involved: (i) splitting the standardized data set into 70% training and 30% testing/validation, and retaining the training data, (ii) using a Nearest Neighbor model ($k = 3$ to $5$) to select data points in the minority class and drawing vectors between neighboring points; and (iii) randomly generating synthetic data along the vectors until the number of records in the minority equal the number of records in the majority.

The training data from the approaches were then used to build the ANN models. The weights and biases of each ANN model were incorporated into equations in C++.

**Suggested intervention.** The intervention was calculated by scoring the predicted averages and standard deviations for successful stone removal and treatment complications. The scoring system was as follows: an average prediction $<0.5$ was scored as 0; a standard deviation $<0.25$ was scored as 0; an average prediction $> = 0.5$ was scored as 1; and a standard deviation that was $> = 0.25$ was scored as 1. The scores for SWL stone removal and treatment complications were summed, as were the scores for URS stone removal and treatment complications. If the sum of SWL was greater than the sum of URS, then the suggested intervention was "URS". If the sum of URS was greater than the sum of SWL, then the suggested intervention was "SWL". If the sum of both SWL and URS were 0, then the suggested intervention was 'SWL or URS". If the sum of SWL and URS was greater or equal to 5 then the suggested intervention was "Uncertain".

## Statistical and data analyses

Averages, standard deviations, and one- or two- tailed Student T-tests were implemented in Excel spreadsheets. One-tailed T-tests were used when direction of the test was relevant and two-tailed T-tests when the direction of the test was unknown. The data was SMOTED using Jupyter notebooks running Python libraries. All ANN models were built and tested using the bench marked Neuroet package downloaded from http://peteranoble.com/software.html. Library (pROC) in the R-program 4.1.2 (2021-11-01) was used to calculate Area-under-the Curve (AUC).

## Results

### Descriptive statistics

The Storz SLX-T instrument was more represented (55.9%) in the SWL data set than the Storz F2 (30.7%) and Dornier instruments (13.4%) (Table 1). Also, more stones were in the kidney (i.e., Lower, Mid, Upper Calyx, Pelvis, and Ureterovesical Junction; 77.8%) than the ureters (Lower, Mid, Upper Ureters and Ureteral Pelvic Junction; 21.6%) or other locations (Bladder, Calcified Stent and Staghorn; <1.0%). Slightly more than half of the patients were overweight healthy males with an average age of 57 years and kidney stones of 8 to 9 mm in diameter.

**Table 1. Descriptive statistics for the SWL data set.**

| Category | Item | SWL data set ($n$ = 15126) |
|---|---|---|
| Instrument used | Dornier Compact Delta II | 6.6% ($n$ = 1003) |
| | Dornier Compact Delta III | 3.5% ($n$ = 536) |
| | Dornier Compact Sigma | 3.1% ($n$ = 472) |
| | Storz F2 | 30.7% ($n$ = 4651) |
| | Storz SLX-T | 56.0% ($n$ = 8464) |
| Stone Location | Ureters | 21.6% (n = 3271) |
| | Kidney | 77.8% ($n$ = 11771) |
| | 0ther locations | <1.0% ($n$ = 84) |
| | Stone side (Left = 0, Right = 1) | 55.4% ($n$ = 8380) |
| Stone properties | Stone Width (mm) | 8.2 ± 4.4 |
| | Stone Length (mm) | 8.7 ± 4.7 |
| Patient Information | Anticoagulants (True = 0; False = 1) | 93.8% ($n$ = 14192) |
| | Gender (Male = 1, Female = 0) | 55.1% (n = 8338) |
| | BMI (kg/m$^2$) | 30.1 ± 6.9 |
| | Age at time of procedure (years) | 57.0 ± 14.9 |
| | Medical Condition (True = 1, False = 0) | 7.8% ($n$ = 1173) |
| Treatment Outcomes | Treatment Complications (False = 0; True = 1) | 4.8% ($n$ = 732) |
| | Stone Removal (Success = 0; Failure = 1) | 15.6% ($n$ = 2353) |

%, proportion in category; $n$, number in category.

Treatment complications were relatively low (<5%) and most kidney stones (84.4%) were successfully removed by SWL.

The Lumenis Versapulse (100 watt and 20 watt) instruments were more represented (63.3%) in the URS data set than other instruments (36.7%) (Table 2). The composition of the patients was similar to those treated by SWL (Table 1) with slightly more than half being overweight males with an average age of 57 years. Treatment complications were relatively low (<5%) and most (92.8%) kidney stones were successfully removed by URS.

**Table 2. Descriptive statistics for URS data.**

| Category | Item | URS data set ($n$ = 2116) |
|---|---|---|
| Instrument used | Dornier Medilas H20 | 26.8% ($n$ = 568) |
| | Dornier Medilas H30 | 3.5% ($n$ = 75) |
| | Dornier Medilas H35 | 2.6% ($n$ = 56) |
| | Lumenis Versapulse 100 watt | 29.3% ($n$ = 621) |
| | Lumenis Versapulse 20 watt | 34.0% ($n$ = 723) |
| | Odyssey Convergent 30 watt | 3.4% ($n$ = 73) |
| Patient Information | | |
| | Gender (Male = 1; Female = 0) | 54% ($n$ = 1142) |
| | Age (years) | 56.5 ± 15.5 |
| | BMI (kg/m$^2$) | 30.4 ± 7.7 |
| Treatment Outcomes | Treatment Complications (False = 0; True = 1) | 5.3% ($n$ = 113) |
| | Stone Removal (Success = 0; Failure = 1) | 7.2% ($n$ = 152) |

%, proportion in category; n, number in category.

**Table 3. Summary of ANN models developed with balanced datasets and tested on the entire data set.** Model accuracies were assessed using a Confusion Matrix and AUC. The Confusion Matrices and AUCs are shown in S1-S8 Tables and S1, S2 Figs in *S1 File*.

| Treatment | Predicted outcome | Model accuracy (%) with balanced data set (70% training: 30% testing/validation) | | Model accuracy (%) with entire SWL data set ($n = 15126$) | | Model accuracy (%) with entire URS data set ($n = 2116$) | |
|---|---|---|---|---|---|---|---|
| | | Confusion matrix | AUC | Confusion matrix | AUC | Confusion matrix | AUC |
| SWL | Stone removal | 73.7 | 77.9 | 22.2 | 50.1 | - | - |
| | Treatment complications | 81.0 | 78.8 | 56.6 | 64.1 | - | - |
| URS | Stone removal | 92.8 | 95.9 | - | | 16.0 | 55.1 |
| | Treatment complications | 80.1 | 78.6 | - | | 63.1 | 51.9 |

## ANN model architecture

Tests of ANN model architectures for the balanced and SMOTED data sets revealed 16 hidden neurons were optimal for SWL models and 5 to 7 hidden neurons were optimal for the URS models.

**Balanced data sets.** Models trained with the balanced data set yielded reasonable prediction accuracies ranging from 72.4 to 92.8% for Confusion Matrices and 77.3 to 95.9% for AUC values (Table 3, S1-S8 Tables and S1, S2 Figs in S1 File). However, when the same models were tested on the entire data sets, model accuracies for Confusion Matrices (balanced data set versus entire data set) were significantly lower (one-tailed T-test, p<0.04). Similar results were obtained for AUC (one-tailed T-test, p<0.01). The presumed reason for these differences is that the minority class was under-represented in the entire data sets. The results demonstrate the need of an alternative approach to improve model predictions, such as modeling using SMOTE approaches.

**SMOTED data sets.** Validation data sets (not used in training or SMOTED) were employed to assess prediction accuracies of the SMOTED models. Table 4 shows that the prediction accuracies based on the Confusion Matrices were reasonable for the SWL and URS models ranging from 82.6% to 93.0%. Interestingly, accuracies based on AUC were sub-optimal, with prediction values ranging from 49.6% to 70.5%. This finding suggests AUCs are more sensitive to the number of minority records (and/or the noise) in the validation data sets than the Confusion Matrices. We will investigate this issue in the next section below.

Comparison of the prediction accuracies of the models (two-tailed T-tests) using the validation data sets and the entire data sets revealed no significant differences for the Confusion Matrix or AUC results (Table 4, S9-S16 Tables and S3, S4 Figs in *S1 File*). The significance of this finding is that models trained with the SMOTED data sets yielded relatively consistent outcomes regardless of the data sets used to test them. Of note, the SMOTED data sets were not used to test the models–they were only used to train the models.

**Table 4. Summary ANN models developed with SMOTED datasets and tested on the validation data set (hold out) and the entire data set.** Model accuracies were assessed using a Confusion Matrix and AUC. The Confusion matrices and AUCs are shown in S9-S16 Tables and S3, S4 Figs in *S1 File*.

| Treatment | Predicted outcome | Model accuracy (%) SWL validation data sets ($n = 4539$) | | Model accuracy (%) using URS validation data sets ($n = 636$) | | Model accuracy (%) with entire SWL data set ($n = 15126$) | | Model accuracy (%) with entire URS data set ($n = 2116$) | |
|---|---|---|---|---|---|---|---|---|---|
| | | Confusion matrix | AUC | Confusion matrix | AUC | Confusion matrix | AUC | Confusion matrix | AUC |
| SWL | Stone removal | 82.6 | 66.9 | - | - | 84.2 | 72.1 | - | - |
| | Treatment complications | 94.0 | 50.7 | - | - | 94.4 | 58.7 | - | - |
| URS | Stone removal | - | - | 89.4 | 64.8 | - | - | 88.6 | 55.1 |
| | Treatment complications | - | - | 88.2 | 49.6 | - | - | 92.6 | 56.4 |

**Table 5. Confusion matrix based on averaged predictions of ten ANN models trained on the SMOTED SWL stone removal data set and tested with the entire data set.**

| Actual (below) /Predictions (across) | 0 | 1 | Sum |
|---|---|---|---|
| 0 | 82.5% (*n* = 12475) | 2.0% (*n* = 298) | 12773 |
| 1 | 13.0% (*n* = 1967) | 2.6% (*n* = 386) | 2353 |
| | | | 85.0% (*n* = 15126) |

0, stone removal success; 1, stone removal failure.

## Predictions using ensembled ANN models

Ensemble processing was used to improve upon model predictions and assess the variability of the predictions of each patient record. This was accomplished by calculating the averages and standard deviations of the predictions from 10 independently SMOTED ANN models. The averaged values were then used to assess model performance (Confusion matrix and AUC).

Model accuracies were 85.0% for SWL stone removal results based on the Confusion Matrix (Table 5) and 74.8% for results based on AUC (Fig 1A). Model accuracy was 95.1% for SWL treatment complication results based on the Confusion Matrix (Table 6) and 66.3% for those based on AUC (Fig 1B).

Model accuracy was 91.2% for URS stone removal results based on the Confusion Matrix (Table 7) and 77.2% for those based on the AUC (Fig 1C), suggesting moderate to high precision. Model accuracy was 93.2% for URS treatment complication results based on the Confusion matrix (Table 8) and 78.9% for results based on AUC (Fig 1D).

The model accuracies for the averaged SMOTED ANN models based on the entire data sets are summarized in Table 9. Two-way T-tests showed no significant differences in predicted outcomes based on Confusion Matrices of individually trained ANN models (Table 4) and those of averaged ANN models (Table 9). However, there were significant improvements in predictions based on AUC results (P<0.027). Specifically, the averaged AUC values increased from 58.0% to 73.4%, suggesting that noise in the data was responsible for the substantially lower AUC results previously reported (Table 4).

In summary, ensemble models improved predictions in two ways: (i) it significantly improved AUC results, and (ii) it enabled Users to access the precision of predictions; those having low standard deviations versus those with high standard deviations, which is important for making intervention decisions of individual patients with kidney stones based on the SDE.

## Assessment of SDE performance

The incorrect SDE predictions could be separated into two categories: (i) those within one standard deviation of the actual value, and (ii) those outside the standard deviation. Incorrect predictions in the first category ranged from 1.1% to 6.1% of the total depending on intervention and outcome, while those in the second ranged from 2.6% to 8.4% (Table 10). Combining the number of correct predictions with the incorrect predictions in the first category revealed that the SDE was reasonably accurate with values ranging from 91.5% to 97.4% (Table 10).

**Individual patients.** Since the SDE was designed to predict outcomes for individual patients, predictions of 10 randomly selected individual patient records were compared to corresponding actual values in the SWL and URS data sets (Table 11) and the suggested intervention was determined.

**SWL stone removal and treatment complications.** All actual values for SWL stone removal indicate that the stones were <4 mm after treatment. The SDE correctly predicted 9 records were <0.5. One of the records was >0.5 but also had a large standard deviation,

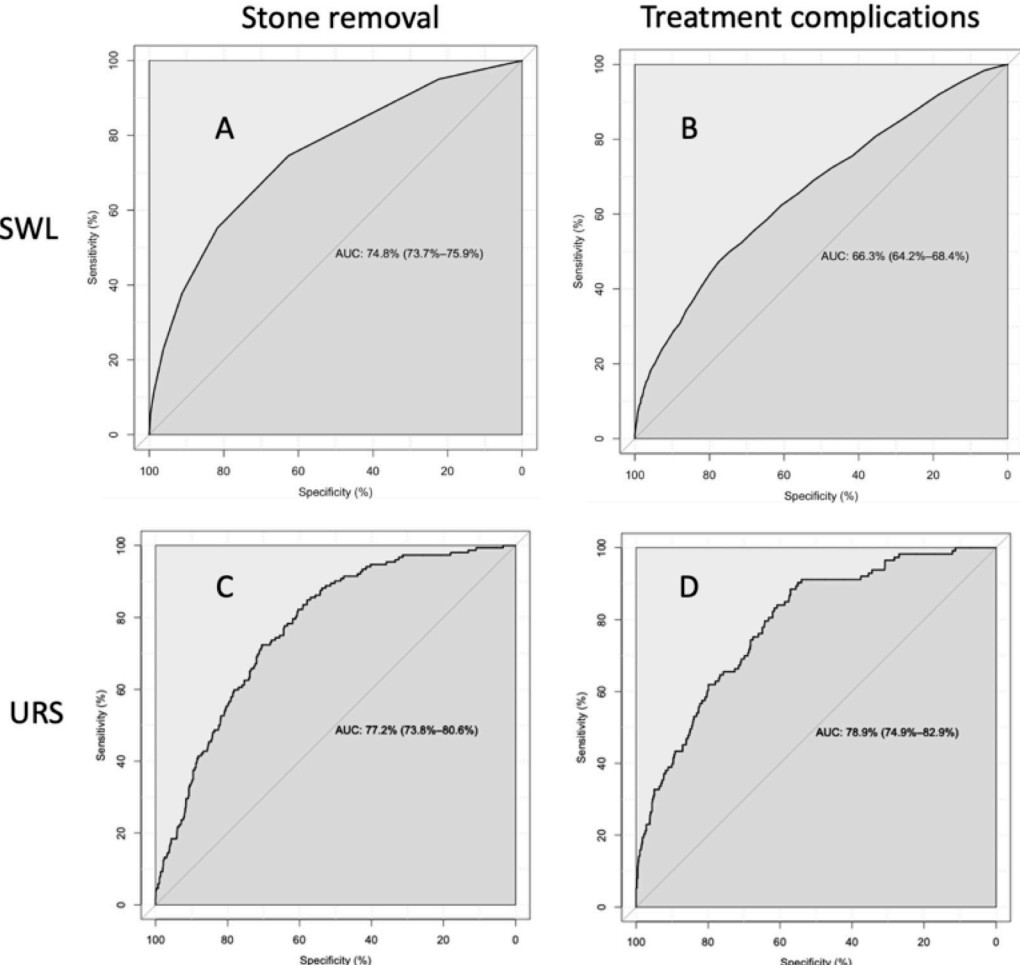

**Fig 1.** AUCs for averaged predictions from ten ANN models trained on SMOTED SWL stone removal data set (A) and treatment complication (B) and tested with the entire data set (*n* = 15126 records). AUC for averaged predictions from ten ANN models trained on SMOTED URS stone removal data set (C) and treatment complication data set (D) and tested with the entire data set (*n* = 2116 records).

indicating the prediction was within one standard deviation of the correct answer (Table 11). The predictions represent 10 of the 13848 records (91.5%) shown in Table 10.

Nine of the 10 actual records for SWL treatment complications were '0', indicating no treatment complications, but one record was '1' indicating a treatment complication (Table 11). The SDE correctly predicted 9 of 10 records but one treatment (i.e., SWL 4) was predicted as a treatment complication with high standard deviation. The significance of this finding is the prediction has high uncertainty but within one standard deviation of the correct answer. The correct predictions are represented as 9 for the 14379 records (95.1%) shown in Table 10 and the uncertain one represents 162 of the 15126 records (1.1%) that are classified as incorrect but within one standard deviation of the correct prediction.

Six of 10 suggested interventions were categorized, as "SWL or URS" because SWL and URS predicted values were <0.5 (Table 11). Three of the suggested interventions were URS (only) because the scoring system showed that SWL was greater than URS. One of the suggested interventions was SWL (only) because the standard deviation of URS treatment complication prediction was >0.25.

**Table 6. Confusion matrix based on averaged predictions of ten ANN models trained on the SMOTED SWL treatment complication data set and tested with the entire data set.**

| Actual (below) /Predictions (across) | 0 | 1 | Sum |
|---|---|---|---|
| 0 | 94.9% ($n = 14355$) | 0.3% ($n = 39$) | 14394 |
| 1 | 4.7% ($n = 708$) | 0.2% ($n = 24$) | 732 |
| | | | 95.1% ($n = 15126$) |

0, no treatment complication; 1, treatment complication.

**Table 7. Confusion matrix based on averaged predictions of ten ANN models trained on the SMOTED URS stone removal data set and tested with the entire data set.**

| Actual (below) /Predictions (across) | 0 | 1 | Sum |
|---|---|---|---|
| 0 | 89.9% ($n = 1902$) | 2.9% ($n = 62$) | 1964 |
| 1 | 5.9% ($n = 125$) | 1.3% ($n = 27$) | 152 |
| | | | 91.2% ($n = 2116$) |

0, successful stone removal; 1, stone removal failure.

**Table 8. Confusion matrix based on averaged predictions of ten ANN models trained on the SMOTED URS treatment complication data set and tested with the entire data set.**

| Actual (below) /Predictions (across) | 0 | 1 | Sum |
|---|---|---|---|
| 0 | 92.2% ($n = 1950$) | 2.5% ($n = 53$) | 2003 |
| 1 | 4.3% ($n = 90$) | 1.1% ($n = 23$) | 113 |
| | | | 93.2% ($n = 2116$) |

0, no treatment complication; 1, treatment complication.

**Table 9. Summary of model accuracies for ensembled SMOTED ANN models ($n = 10$) tested with entire data sets.** Model accuracies were assessed using Confusion Matrix and AUC.

| Treatment | Predicted outcome | Model accuracy (%) with entire SWL data set ($n = 15126$) | | Model accuracy (%) with entire URS data set ($n = 2116$) | |
|---|---|---|---|---|---|
| | | Confusion matrix | AUC | Confusion matrix | AUC |
| SWL | Stone removal | 85.0 | 74.8 | - | - |
| | Treatment complications | 95.0 | 66.3 | - | - |
| URS | Stone removal | - | - | 91.2 | 77.2 |
| | Treatment complications | - | - | 93.2 | 78.9 |

**URS stone removal and treatment complications.** Nine of the 10 actual records for URS stone removal were '0', indicating successful stone removal, but one record was '1', indicating that the stone was >4mm after treatment (Table 11). The SDE correctly predicted 8 of the 10 records. One of the two incorrectly predicted records had a high standard deviation (20%) indicating that the prediction was within one standard deviation of the correct value. This record represents one of the 130 (6.1%) shown in Table 10. The other record was a false negative (in bold) and represents one of the 102 records (4.8%) in Table 10.

All ten actual records for URS treatment complications were '0', indicating no treatment complications and the SDE correctly predicted these records (Table 11). These predictions represent 10 of the 1950 records (92.2%) shown in Table 10.

**Table 10. Prediction performance of SDE (40 model equations) by intervention and outcome.**

| Intervention (across) | SWL (n = 15126 records) | | URS (n = 2116 records) | |
|---|---|---|---|---|
| Outcome | Stone removal | Treatment complications | Stone removal | Treatment complications |
| Correct predictions | 85.0% (n = 12861) | 95.1% (n = 14379) | 89.0% (n = 1884) | 92.2% (n = 1950) |
| Incorrect predictions but within STD | 6.5% (n = 987) | 1.1% (n = 162) | 6.1% (n = 130) | 5.1% (n = 110) |
| Incorrect Prediction | 8.4% (n = 1278) | 3.9% (n = 585) | 4.8% (n = 102) | 2.6% (n = 56) |
| Correct predictions and/or incorrect predictions within STD | 91.5% | 96.2% | 95.1% | 97.3% |

Six of 10 suggested interventions were categorized, as "SWL or URS" because SWL and URS predicted values were <0.5 (Table 11). Three records were categorized as SWL (only) because the scoring system found that URS > SWL. One record was categorized as 'URS' because the scoring system found that URS < SWL.

In summary, the SDE demonstrated reasonable accuracy in predicting outcomes based on patient information. To aid healthcare providers in counseling patients and determining the optimal treatment options for stones in the urinary tract, we have developed a user-friendly SDE web interface, which can be accessed at http://peteranoble.com/webapps.html.

## Discussion

The primary motivation of our study was driven by the desire to provide healthcare professionals with a data-driven tool to accurately predict treatment outcomes based on patient information and intervention (SWL and URS). To our knowledge, this is the first large-scale

**Table 11. Ten random selected examples of the prediction performance of SDE by intervention, outcome and suggested intervention.**

| Intervention by individual patient | Actual stone removal (SR) (0 = Success; 1 = Failure) | Predicted SR ± Stdev | Actual treat complications (TC) (False = 0; True = 1) | Predicted TC ± Stdev | Suggested Intervention |
|---|---|---|---|---|---|
| SWL 1 | 0 | 0.05 ± 0.03 | 0 | 0.12 ± 0.06 | SWL_or_URS |
| SWL 2 | 0 | 0.32 ± 0.28 | 1 | 0.68 ± 0.22 | URS |
| SWL 3 | 0 | 0.06 ± 0.18 | 0 | 0.05 ± 0.16 | SWL_or_URS |
| SWL 4 | 0 | 0.48 ± 0.37 | 0 | 0.70 ± 0.42* | URS |
| SWL 5 | 0 | 0.26 ± 0.24 | 0 | 0.18 ± 0.23 | SWL_or_URS |
| SWL 6 | 0 | 0.06 ± 0.17 | 0 | 0.06 ± 0.17 | SWL_or_URS |
| SWL 7 | 0 | 0.01 ± 0.14 | 0 | 0.28 ± 0.35 | SWL_or_URS |
| SWL 8 | 0 | 0.03 ± 0.13 | 0 | 0.01 ± 0.20 | SWL |
| SWL 9 | 0 | 0.07 ± 0.13 | 0 | 0.06 ± 0.16 | SWL_or_URS |
| SWL 10 | 0 | 0.68 ± 0.39* | 0 | 0.07 ± 0.21 | URS |
| URS 1 | 0 | 0.00 ± 0.14 | 0 | 0.18 ± 0.29 | URS |
| URS 2 | 0 | **0.76 ± 0.24** | 0 | 0.02 ± 0.18 | SWL |
| URS 3 | 0 | 0.10 ± 0.18 | 0 | 0.01 ± 0.15 | SWL_or_URS |
| URS 4 | 0 | 0.00 ± 0.14 | 0 | 0.01 ± 0.13 | SWL_or_URS |
| URS 5 | 0 | 0.00 ± 0.13 | 0 | 0.00 ± 0.12 | SWL_or_URS |
| URS 6 | 0 | 0.47 ± 0.20 | 0 | 0.04 ± 0.14 | SWL_or_URS |
| URS 7 | 0 | 0.00 ± 0.15 | 0 | 0.00 ± 0.12 | SWL_or_URS |
| URS 8 | 1 | 0.37 ± 0.20* | 0 | 0.27 ± 0.29 | SWL |
| URS 9 | 0 | 0.44 ± 0.20 | 0 | 0.27 ± 0.29 | SWL |
| URS 10 | 0 | 0.19 ± 0.20 | 0 | 0.01 ± 0.18 | SWL_or_URS |

*, Incorrect prediction but within standard deviation (Stdev); Bold, incorrect prediction.

study to predict stone treatment outcomes using ANN modeling. Our study is unique from other studies because the interventions took place at multiple institutions ($n = 41+$) by different medical professionals ($n = 41+$) using a variety of SWL and URS instruments (Tables 1 and 2). Hence, the results should be generalizable and not specific to a particular institution, healthcare professional, or instrument. While there are specific guidelines for the management of urolithiasis set by the American Urological Association (AUA) and European Association of Urologists (EAU), our study provides recommendations based on past treatments that in theory should align with these guidelines.

The secondary motivation was to demonstrate the utility of ANN models to solve complex healthcare problems. Our initial studies using balanced data sets yielded sub-optimal results (Table 3), presumably due to the minority class biasing the predictions when tested with the entire data sets. The SMOTED data substantially increased the representation of the minority class and consequently improved predictions, as shown in this study and others [33–36]. Ensembling by averaging the predictions of multiple diverse models reduced the error and improved upon the final predictions (compare Tables 4 to 9). The diverse models in our study were due to different random splits of the data, randomization of the SMOTE process, and randomization of the initial sets of weights and biases of the ANN models prior to training. Previous studies have used ensemble processes to improve predictions over those made by individually trained models [37, 38]. An additional advantage of the ensemble process in our study was that the variability of the predictions for individual patient records could be determined.

The strengths of our study are that the models were based on 17242 patients–far more than other studies; and the predictions should be generalizable because the data were collected from many different institutions, with different healthcare professionals, and a variety of SWL and URS instruments. One limitation of our study is its retrospective design, which may have led to biases and reduced the predictive accuracies of the ensembled models. Ongoing prospective studies may improve upon our findings.

## Model predictions based on confusion matrix versus those on AUC

We investigated prediction accuracies using Confusion Matrices and AUCs to highlight similarities and differences of the two assessment approaches.

A Confusion Matrix measures the performance of a classifier using a fixed threshold. Predictions <0.5, for example, were classified as '0', which corresponds to either 'successful stone removal' or 'no treatment complications', and predictions >0.5 were classified as '1', which corresponds to 'stone removal failure' or 'treatment complications'. The accuracy of a model was defined by the sum of the True Positives and True Negatives divided by the total number of samples and reported as a percent.

In contrast, AUC examines the performance of a classifier without any fixed threshold—every possible threshold is examined and plotted as a point on the curve—and it is reported as a percent. The two approaches differ because AUC is apparently more sensitive to noise in the data than the Confusion Matrix, as demonstrated in this study by the improvement of AUC values after multiple independent predictions were ensembled.

## Input variables to the SDE

Previous studies have shown SWL variables affecting treatment outcomes include gender [39–42], age [39, 42–44], SSD [40, 45–50], BMI [39, 50], stone number [39, 42, 43], stone size [39–43, 46, 47, 50–56], stone location [39, 41–43, 48, 51, 52, 56, 57], and stone characteristics [33–

48, 51, 54–58]. Variables affecting URS outcomes include stone number [59, 60], stone size [53, 59], stone location [59, 60], and stone characteristics [59, 60].

While some overlap exists in the variables affecting SWL and URS outcomes, more variables have been shown to affect SWL outcomes than URS outcomes. These differences were considered during the construction of the SDE and explain the different number of input variables used to predict SWL and URS outcomes in our study. The choice of input variables was also dependent on the number of missing or erroneous values (e.g., BMI of >100) in the data sets since rows and columns containing numerous missing or erroneous values were excluded from the study.

## Comparison to other studies in the literature

Nomograms and mathematical models have been used to predict SWL and URS outcomes in many previous studies. Nomograms are graphical decision-making tools that are easy to use, and they do not require knowledge of the underlying equation that the nomogram represents. Predictive mathematical models consist of coefficients that are multiplied by input variables and summed to yield a predictive outcome. ANN models fall into this category with the coefficients being the weights and biases of the trained network.

Here, we briefly document previous studies (in chronological order by intervention) and where appropriate, mention their limitations.

**SWL studies.** Kanao et al. [61] created one of the first nomograms to predict stone-free rates based on 435 patients. While the nomogram considered stone size, stone location, and stone number, critics argued that their approach was inadequate [49] because it did not consider stone density and skin-to-stone distance (SSD).

Vakalopoulos et al. [39] constructed a mathematical model predicting the successful outcomes of 1712 patients. The approach was unique from others because the equations were presented. The stated limitations are: (i) different stone locations (i.e., renal, ureter, and total) required different models; (ii) the models would have to be adjusted for different lithotripters; and (iii), the model needed to be validated prospectively to prove its usefulness.

Two studies developed nomograms predicting SWL stone-free rates in children. The Onal et al. [62] model was based on 395 patients. The limitation of the study was that the model was based on one urologist at a single institution, and a single instrument and the approach has not been externally validated. The Dongan and Tekgul [63] predicted stone-free rates and complication rates. Yanaral et al. [64] argued that both Onal et al. [62] and Dongan and Tekgul [63] studies could be improved by the addition of variables such as stone density, degree of obstruction, shock power, and number of shocks applied.

Wiesenthal et al. [65] examined 422 patients to find that predictors of successful lithotripsy differed by stone location and therefore developed two mathematical equations: one for the kidney and the other for the ureter. The stated limitations are that the models did not consider the different types of lithotripters, nor did they include a diversity of institutions and operators.

Tran et al. [58] developed the Triple D score to predict stone-free rates in 235 patients. The model was developed by applying threshold values to AUC curves for ellipsoid stone volume, SSD, and stone density. The score was based on the sum of the number of parameters that fell below the thresholds. The research has been validated by Ichiyanagi et al. [66] with 226 patients.

Kim et al. [57] predicted stone-free rates for 3028 patients from three independent institutions and developed a nomogram based on sex, stone location, stone number, stone size, mean

Hounsfield unit and grade of hydronephrosis. The model could also be used to advise patients on the likelihood of single or multiple SWL treatments.

Ickiyanagi et al. [66] developed the Quadruple D score based on 226 patients to predict renal stone free status. The scoring system was defined as the sum of the Triple D score [56] and a number based on stone location in the kidney. The stated limitations were: (i) the score did not consider stone morphology or hydronephrosis grade; (ii) the score was not tested on stones in the ureters; (iii) the study had limited diversity as it was based on Japanese patients; and (iv) it has not been externally validated.

Yoshioka et al. [56] developed an integer score-based prediction model (S3HoCKwave score) for assessing SWL failure based on 2271 patients. The study was conducted at several medical centers and was shown to be superior to Triple D score developed by Tran et al. [58]. In the model, continuous outcomes were converted to dichotomous outcomes, and then multivariable logistic regression analysis calculated the coefficients for each prediction. The values of each prediction were rounded, multiplied by 10 and summed. Assessment of performance was based on internal and external validation. The stated limitations are that the study was based on Asian population and limited to non-contrast-enhanced computed tomography.

**URS studies.** Resorlu et al. [59] developed a scoring system to predict stone free status based on 207 patients using the following variables: stone size, composition, stone number, renal malformation and lower pole infundibulopelvic angle. Each variable (excluding composition) was scored as either zero or one based on yes or no answers. While the system was limited to a few patients, it has been externally verified by Wang et al. [67] and Bozkurt et al. [68].

Imamura et al. [69] developed a nomogram based on 412 patients that predicted stone free rate. De Nunzi et al. [70] validated the Imamura nomogram using 275 European patients.

Jung et al. [71] developed a modified S-ReSC score based on 88 patients to predict stone free status; but the low number of patients limits the usefulness of the score although it has been externally evaluated [68].

Ito et al. [60] develop a scoring system for stone free status based on 310 patients using stone volume, stone location, operator experience, stone number and presence of hydronephrosis. The score was derived by the sum of individual scores. The stated limitation of the system is too few patients but it has been externally evaluated [68].

Xiao et al. [72] developed the R.I.R.S system based on 382 patients to predict stone free status of 4 parameters: renal stone density, inferior pole stone, renal infundibular length and cumulative stone diameter. It has been externally evaluated [68].

Bozkurt et al. [68] examined four of the five URS nomograms mentioned above [i.e., 59, 60, 71, 72] with 949 patients from two institutions. While the nomograms predicted stone free status and treatment complications with varying degrees of success, Bozkurt stated that the nomograms have limitations, and an ideal system has yet to be developed.

**Nomogram for SWL, retrograde intrarenal surgery (RIRS), and percutaneous nephrolithotomy (PNL) interventions.** Micali et al. [73] develop a nomogram for predicting treatment failure of solitary kidney stones between 1 and 2 cm in size for SWL, RIRS and PNL. The input data for their model was preoperative clinical data. They stated that external validation of the current nomogram was needed to determine its reproducibility and validity.

## Conclusions

This is the first large-scale multi-site study to develop a SDE that accurately predicts SWL and URS outcomes for prospective patients. A practical outcome of this research is a SDE web interface that can help healthcare providers in counseling patients and determining the optimal treatment options: http://peteranoble.com/webapps.html.

## Supporting information

**S1 File. The file 'Supplementary Materials.docx' contains 16 tables and 4 figures.**
(DOCX)

## Acknowledgments

We thank Dr. Alex Pozhitkov from the City of Hope Cancer Research Center for his critical comments in an earlier version of the manuscript and testing the SDE web interface. We also thank Derek Soetemans and Ifeanyi Okwuchi from the Focus21 team for their help in critical improvements in earlier versions of the manuscript.

## Author Contributions

**Conceptualization:** Peter A. Noble.

**Formal analysis:** Peter A. Noble.

**Investigation:** Peter A. Noble.

**Methodology:** Peter A. Noble.

**Project administration:** Peter A. Noble.

**Software:** Peter A. Noble.

**Supervision:** Peter A. Noble, Blake D. Hamilton, Glenn Gerber.

**Validation:** Peter A. Noble.

**Visualization:** Peter A. Noble.

**Writing – original draft:** Peter A. Noble.

**Writing – review & editing:** Peter A. Noble, Blake D. Hamilton, Glenn Gerber.

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
