## [Decision Letter · Decision Letter 0]

5 Jan 2024

PONE-D-23-38798Kidney Stone Decision Engine accurately predicts kidney stone removal and treatment complications for shock wave lithotripsy and laser ureterorenoscopy patientsPLOS ONE

Dear Dr. Noble,

Thank you for submitting your manuscript to PLOS ONE. After careful consideration, we feel that it has merit but does not fully meet PLOS ONE’s publication criteria as it currently stands. Therefore, we invite you to submit a revised version of the manuscript that addresses the points raised during the review process.

We look forward to receiving your revised manuscript.

Kind regards,

Ahmed Abdelmotteleb Taha Eissa, Ph.D., M.D.

Academic Editor

PLOS ONE

Journal Requirements:

5. Please note that in order to use the direct billing option the corresponding author must be affiliated with the chosen institute. Please either amend your manuscript to change the affiliation or corresponding author, or email us at plosone@plos.org with a request to remove this option.

7. Your ethics statement should only appear in the Methods section of your manuscript. If your ethics statement is written in any section besides the Methods, please move it to the Methods section and delete it from any other section. Please ensure that your ethics statement is included in your manuscript, as the ethics statement entered into the online submission form will not be published alongside your manuscript.

Additional Editor Comments:

**ACADEMIC EDITOR:**

In the submission process the authors indicated that the article was not funded; however, in the manuscript itself, they stated that they received funding from Translational Analytics and Statistics, please change the funding status in the submission.

Reviewers' comments:

Reviewer's Responses to Questions

**Comments to the Author**

1. Is the manuscript technically sound, and do the data support the conclusions?

Reviewer #1: Partly

Reviewer #2: Yes

2. Has the statistical analysis been performed appropriately and rigorously? 

Reviewer #1: I Don't Know

Reviewer #2: Yes

3. Have the authors made all data underlying the findings in their manuscript fully available?

Reviewer #1: No

Reviewer #2: No

4. Is the manuscript presented in an intelligible fashion and written in standard English?

Reviewer #1: Yes

Reviewer #2: Yes

5. Review Comments to the Author

Reviewer #1: This study aimed to predict successful stone removal and complication outcomes using Artificial Neural Network models on 15,126 SWL and 2,116 URS patient records. the idea is interesting but I have one comment below:

1. The complications outcomes were put all together and this should have been classified according to the modified Clavien classification for reporting complications

Thanks for giving me the opportunity to review your good work!

Reviewer #2: The authors present an interesting article about the use of ANN model to predict outcomes of URS+Laser and SWL in the management of different types of stones; however, I have some concerns that needs to be addressed:

1- The title is not accurate because the authors indicates only kidney stones; while in the methods they included ureteral stones so this should be modified to reflect better the idea of the manuscript.

2- In the introduction section the authors are speaking broadly about different stone sizes and locations including different renal and ureteral stones; however, according to the EAU guidelines this is not totally sound as for example renal stones > 2cm in size should be treated initially by PCNL, and in some situations URS is the preferred option over SWL in distal ureteral stones >10 mm in size. So I think the authors should discuss why should a surgeon refer to such model in cases where there are clear guidelines

3- I think the authors should discuss more the use of Artificial Intelligence in urolithiasis in the introduction section.

4- The used data is not completely reported for example the authors reported that kidney stones represents 77.8% of the cases without identification of the accurate location of the stones; this makes a difference specially in the lower calyceal stones; similarly, the ureteral stones are not classified into proximal or distal (or upper, middle, and lower).. This should be clearly reported in the study. Furthermore, why include a bladder stone???? SWL and URS are not the best treatment options for bladder stones??? And in other locations you mentioned Pancreas????

5- The complications are also reported broadly without differentiation; the authors should indicate the complications more precisely using the Clavien Dindo classification.

6- I didn't understand how was the variable chosen for the model; was it based on other publication or something else.

7- The limitations of the study should be moved from the conclusion to the discussion section.

8- There is a similar nomogram (PMID: 33419709) that is used to predict the outcomes of PCNL, RIRS, or ESWL in single 1-2 cm sized renal stone to help surgeons chose the best decision; I think it should be included in the discussion with other nomograms

6. PLOS authors have the option to publish the peer review history of their article (what does this mean?). If published, this will include your full peer review and any attached files.

Reviewer #1: No

Reviewer #2: No

---

## [Author Response · Author response to Decision Letter 0]

17 Jan 2024

Responses to referees and changes are in red.

Journal Requirements:

Response: Done

Response: The manuscript provides explicit details on how the final program was produced using Jupyter notebooks, Neuroet, and C++ code. The final code is not publically available because it is partially owned by a third party. 

However, I have provided the final product as a user-friendly web interface where people can input their own data. My program will automatically make the calculations, and output figures with recommendations (http://peteranoble.com/webapps.html).

Response: Done

Response: I was not funded by any grant for this research. However, the company ‘Translational Analytics and Statistics’ employed me. I wrote the manuscript and made the web page interface after I left the company.

I changed: 

Funding. None.

5. Please note that in order to use the direct billing option the corresponding author must be affiliated with the chosen institute. Please either amend your manuscript to change the affiliation or corresponding author, or email us at plosone@plos.org with a request to remove this option.

Response: I am a retired adjunct professor and am no longer officially associated with University of Alabama. I have not received funding for ten years. Therefore, the direct billing option is not an option. 

One of my co-authors (Gerber) is from the University of Chicago and it is my understanding that publishing in PlosOne is free for that university. In addition, I will email plosone@plos.org with a request to remove this option.

Response: Understood. Thanks for pointing out multiple errors. The ‘data not shown’ should not have been there in the first place. The statement should be Table 11. I fixed it in multiple places.

7. Your ethics statement should only appear in the Methods section of your manuscript. If your ethics statement is written in any section besides the Methods, please move it to the Methods section and delete it from any other section. Please ensure that your ethics statement is included in your manuscript, as the ethics statement entered into the online submission form will not be published alongside your manuscript.

Response: I put the following ethic statement at the start of the Material and Methods section.

Ethics statement The research relied on the analysis of anonymized data accessible through the Kidney Stone Registry. The anonymous dataset lacks identifiable information, ensuring no possible linkage to personal data.

Additional Editor Comments:

ACADEMIC EDITOR:

In the submission process the authors indicated that the article was not funded; however, in the manuscript itself, they stated that they received funding from Translational Analytics and Statistics, please change the funding status in the submission.

Response: I was not funded by any grant for this research. However, the company ‘Translational Analytics and Statistics’ employed me. I wrote the manuscript and made the web page interface after I left the company.

I changed: 

Funding. None.

Specific comments to the Author

1. Is the manuscript technically sound, and do the data support the conclusions?

Reviewer #1: Partly

Reviewer #2: Yes

Response: Thank you. I hope my changes have sufficiently improved the manuscript to change Reviewer #1’s comment from Partly to Yes.

2. Has the statistical analysis been performed appropriately and rigorously?

Reviewer #1: I Don't Know

Reviewer #2: Yes

Response: Thank you. I hope my changes have sufficiently improved the manuscript to change Reviewer #1’s comment from Partly to Yes.

3. Have the authors made all data underlying the findings in their manuscript fully available?

Response: 

1. I understand the PLOS Data policy. I have no control over access to the dataset used in this study as I do not own it. It is owned by the Kidney Stone Registry and my understanding is any scientist can contact them through the email contact info@trans-stat.com to inquire upon the availability. 

2. The summary statistics presented in Table 11 are a crude example of ten randomly selected examples for demonstration purposes. In the web-interface (http://peteranoble.com/webapps.html), Users can enter their data and summary statistics are presented in the form of a histogram with mean and standard deviations specific to the patient’s input. The User interface is the final product of the model.

Reviewer #1: No

Reviewer #2: No

4. Is the manuscript presented in an intelligible fashion and written in standard English?

Reviewer #1: Yes

Reviewer #2: Yes

Response: Thank you.

5. Review Comments to the Author

Reviewer #1: This study aimed to predict successful stone removal and complication outcomes using Artificial Neural Network models on 15,126 SWL and 2,116 URS patient records. the idea is interesting but I have one comment below:

1. The complications outcomes were put all together and this should have been classified according to the modified Clavien classification for reporting complications.

Response: I understand the desire to classify according to the modified Clavien classification for reporting complications. I wish I had that data. However, the anonymized dataset did not contain that information. Moreover, the request is beyond the original scope of the project.

Thanks for giving me the opportunity to review your good work!

Response: I appreciate your willingness and time spent to review my manuscript. Thank you.

Reviewer #2: The authors present an interesting article about the use of ANN model to predict outcomes of URS+Laser and SWL in the management of different types of stones; however, I have some concerns that needs to be addressed:

1- The title is not accurate because the authors indicates only kidney stones; while in the methods they included ureteral stones so this should be modified to reflect better the idea of the manuscript.

Response: 

1. Thanks for your willingness and time spent to review my manuscript.

2. Agreed. The new title is:

Stone Decision Engine accurately predicts stone removal and treatment complications for shock wave lithotripsy and laser ureterorenoscopy patients

3. I prefer not to include kidney, ureter (and bladder) stones in the title because it would complicate the title and there would be too many words. 

I hope my ‘fix’ to the title sufficiently satisfies the reviewer’s comment.

2- In the introduction section the authors are speaking broadly about different stone sizes and locations including different renal and ureteral stones; however, according to the EAU guidelines this is not totally sound as for example renal stones > 2cm in size should be treated initially by PCNL, and in some situations URS is the preferred option over SWL in distal ureteral stones >10 mm in size. So I think the authors should discuss why should a surgeon refer to such model in cases where there are clear guidelines.

Response: Agreed. We inserted the following statement in the first paragraph of the discussion. 

While there are specific guidelines for the management of urolithiasis set by the American Urological Association (AUA) and European Association of Urologists (EAU), our study provides recommendations based on past treatments that in theory should align with these guidelines.

3- I think the authors should discuss more the use of Artificial Intelligence in urolithiasis in the introduction section.

Response: Agreed. The following section has been added to the Introduction.

Artificial neural network (ANN) models are computational systems or algorithms designed to simulate human intelligence and perform tasks that typically require human intelligence. These models learn from data and experience, enabling them to make predictions, recognize patterns, and solve problems without being explicitly programmed for each specific task. They are now widely used in urology to detect kidney stones in videos [18] and images [19-24], predict sepsis risk [25,26] and lithotripsy treatment outcomes [27-29], and set SWL machine parameters [30].

4- The used data is not completely reported for example the authors reported that kidney stones represents 77.8% of the cases without identification of the accurate location of the stones; this makes a difference specially in the lower calyceal stones; similarly, the ureteral stones are not classified into proximal or distal (or upper, middle, and lower).. This should be clearly reported in the study. Furthermore, why include a bladder stone???? SWL and URS are not the best treatment options for bladder stones??? And in other locations you mentioned Pancreas????

Response: The reviewer's request extends beyond the intended scope of the study. Our primary objective was to develop a Kidney Stone Decision Engine based on a sample of 17,242 treatment cases. Our focus was not specific stone locations but rather general stone locations, aiming to answer the question: Is it possible to predict stone removal and treatment complications based on patient data?

While we acknowledge the importance of specific stone locations, our decision to omit them in this study was deliberate. We aimed to create a generalizable model. Moreover, including specific locations might have led to lower R2 and AUC values in our model predictions, making them less informative. Nonetheless, our study represents a significant step towards achieving predictable outcomes, and perhaps we might investigate specific stone locations in the future.

Our dataset includes bladder stones, which are treated by both Shock Wave Lithotripsy (SWL) and Ureteroscopy (URS). We did not make any assertions about the optimal treatment options for bladder stones.

Pancreas samples were excluded from the dataset prior to analysis and their appearance in the paper is an error. Hence, the term 'pancreas' has been removed from the manuscript.

5- The complications are also reported broadly without differentiation; the authors should indicate the complications more precisely using the Clavien Dindo classification.

Response: The data set did not contain information on the Clavien Dindo classifications. Treatment complications were in binary format: yes or no. 

6- I didn't understand how was the variable chosen for the model; was it based on other publication or something else.

Response: The variables chosen for the model were based on the provided data as well as input variables from previous studies as stated in the ‘Input variables to the SDE section’. 

7- The limitations of the study should be moved from the conclusion to the discussion section.

Response: Agreed. The limitations of the study were moved to the discussion. 

8- There is a similar nomogram (PMID: 33419709) that is used to predict the outcomes of PCNL, RIRS, or ESWL in single 1-2 cm sized renal stone to help surgeons chose the best decision; I think it should be included in the discussion with other nomograms

Response: Agreed. I inserted the following into the manuscript and added the reference to the reference section.

Nomogram for SWL, retrograde intrarenal surgery (RIRS), and percutaneous nephrolithotomy (PNL) interventions

Micali et al. [73] develop a nomogram for predicting treatment failure of solitary kidney stones between 1 and 2 cm in size for SWL, RIRS and PNL. The input data for their model was preoperative clinical data. They stated that external validation of the current nomogram was needed to determine its reproducibility and validity.

---

## [Decision Letter · Decision Letter 1]

25 Mar 2024

Stone Decision Engine accurately predicts stone removal and treatment complications for shock wave lithotripsy and laser ureterorenoscopy patients

PONE-D-23-38798R1

Dear Dr. Noble,

We’re pleased to inform you that your manuscript has been judged scientifically suitable for publication and will be formally accepted for publication once it meets all outstanding technical requirements.

Kind regards,

Ahmed Abdelmotteleb Taha Eissa, Ph.D., M.D.

Academic Editor

PLOS ONE

Additional Editor Comments (optional):

Thank you for performing the required modifications

Reviewers' comments:

Reviewer's Responses to Questions

**Comments to the Author**

1. If the authors have adequately addressed your comments raised in a previous round of review and you feel that this manuscript is now acceptable for publication, you may indicate that here to bypass the “Comments to the Author” section, enter your conflict of interest statement in the “Confidential to Editor” section, and submit your "Accept" recommendation.

Reviewer #2: All comments have been addressed

2. Is the manuscript technically sound, and do the data support the conclusions?

Reviewer #2: Yes

3. Has the statistical analysis been performed appropriately and rigorously? 

Reviewer #2: I Don't Know

4. Have the authors made all data underlying the findings in their manuscript fully available?

Reviewer #2: No

5. Is the manuscript presented in an intelligible fashion and written in standard English?

Reviewer #2: Yes

6. Review Comments to the Author

Reviewer #2: The authors has responded well to the comments, I would like to congratulate them for their efforts

7. PLOS authors have the option to publish the peer review history of their article (what does this mean?). If published, this will include your full peer review and any attached files.

Reviewer #2: No
